# Urinary polycyclic aromatic hydrocarbon metabolites and mortality in the United States: A prospective analysis

Achal P. Patel[1]☯, Suril S. Mehta[2]☯*, Alexandra J. White[3], Nicole M. Niehoff[3], Whitney D. Arroyave[4], Amy Wang[2], Ruth M. Lunn[2]

**1** Department of Epidemiology, University of North Carolina, Chapel Hill, Chapel Hill, NC, United States of America, **2** Division of the National Toxicology Program, Office of the Report on Carcinogens, National Institute of Environmental Health Sciences, Research Triangle Park, NC, United States of America, **3** Epidemiology Branch, National Institute of Environmental Health Sciences, Research Triangle Park, NC, United States of America, **4** Integrated Laboratory Systems, Morrisville, NC, United States of America

☯ These authors contributed equally to this work.
\* suril.mehta@nih.gov

**Data Availability Statement:** All NHANES and NDI files are available for download from the National Center for Health Statistics database (https://www.cdc.gov/nchs/nhanes/index.htm).

## Abstract

### Background

Polycyclic aromatic hydrocarbons (PAHs) are ubiquitous organic compounds associated with chronic disease in epidemiologic studies, though the contribution of PAH exposure on fatal outcomes in the U.S. is largely unknown.

### Objectives

We investigated urinary hydroxylated PAH metabolites (OH-PAHs) with all-cause and cause-specific mortality in a representative sample of the U.S. population.

### Methods

Study participants were $\geq$20 years old from the National Health and Nutrition Examination Survey 2001–2014. Concentrations (nmol/L) of eight OH-PAHs from four parent PAHs (naphthalene, fluorene, phenanthrene, pyrene) were measured in spot urine samples at examination. We identified all-cause, cancer-specific, and cardiovascular-specific deaths through 2015 using the National Death Index. We used Cox proportional hazards regression to estimate adjusted hazard ratios (HRs) and 95% confidence intervals (CIs) for the association between ΣOH-PAHs and mortality endpoints. We assessed potential heterogeneity by age, gender, smoking status, poverty, and race/ethnicity. Additionally, we examined the overall mixture effect using quantile g-computation.

### Results

In 9,739 eligible participants, there were 934 all-cause deaths, 159 cancer-specific deaths, and 108 cardiovascular-specific deaths (median 6.75 years follow-up). A $\log_{10}$ increase in ΣOH-PAHs was associated with higher all-cause mortality (HR$_{adj}$ = 1.39 [95%CI: 1.21,

**Funding:** S.S.M., A.W., R.M.L., National Institute of Environmental Health Sciences, Intramural Research Program Project ES-103317-05, The funders had no role in study design, data collection and analysis, decision to publish, or preparation of the manuscript. A.P.P., National Institute of Environmental Health Sciences, Training Grant T32 ES007018, The funders had no role in study design, data collection and analysis, decision to publish, or preparation of the manuscript.

**Competing interests:** The authors have declared that no competing interests exist.

1.61]), and possibly cancer-specific mortality ($HR_{adj}$ = 1.15 [95%CI: 0.79, 1.69]), and cardiovascular-specific mortality ($HR_{adj}$ = 1.49 [95%CI: 0.94, 2.33]). We observed substantial effect modification by age, smoking status, gender, and race/ethnicity across mortality endpoints. Risk of cardiovascular mortality was higher for non-Hispanic blacks and those in poverty, indicating potential disparities. Quantile g-computation joint associations for a simultaneous quartile increase in OH-PAHs were $HR_{adj}$ = 1.15 [95%CI: 1.02, 1.31], $HR_{adj}$ = 1.41 [95%CI: 1.05, 1.90], and $HR_{adj}$ = 0.98 [95%CI: 0.66, 1.47] for all-cause, cancer-specific, and cardiovascular-specific mortalities, respectively.

## Discussion

Our results support a role for total PAH exposure in all-cause and cause-specific mortality in the U.S. population.

## Introduction

Polycyclic aromatic hydrocarbons (PAHs) are organic compounds with fused aromatic rings commonly produced via incomplete combustion of organic material. Major PAH sources in the United States include vehicle exhaust, tobacco smoke, coal tar, grilled and smoked foods, agricultural burning, and occupational sources [1]. PAHs are widely distributed in the environment and exposure to the U.S. population is ubiquitous, generally as a complex mixture of multiple PAHs [2, 3].

In human observational studies, PAH exposure has been associated with a range of chronic, and often fatal, diseases including ischemic heart disease, impaired respiratory function, lung cancer, and breast cancer [4–9]. Occupations with high levels of PAH exposure, including coke ovens, aluminum production, asphalt, and chimney sweeping are associated with excess mortality from lung and other cancers, cardiovascular diseases, and non-malignant respiratory diseases [4, 5, 10–14]. The International Agency for Research on Cancer (IARC) [15] and the U.S. National Toxicology Program's Report on Carcinogens (RoC) [16] have classified more than a dozen individual PAHs as known or likely carcinogens primarily based on mechanistic and toxicological data.

Methods for assessing exposure to PAHs include job-exposure matrices in occupational settings, exposure modeling, personal air monitor sampling, self-reported exposures, and biomarkers [5, 7–9]. Biomarkers reflect exposure from across multiple routes and thus may better represent internal or biological dose compared to methods that consider a specific exposure route (e.g., personal air sampling). PAH biomarkers reflect shorter term exposure, with a half-life of a week to months for PAH-DNA adducts and hours to days for urinary hydroxylated (OH) PAH metabolites (OH-PAHs) [17–19]. Genetic polymorphisms in activation and detoxification pathways impact levels of both types of PAHs biomarkers, while PAH-DNA adduct levels are additionally influenced by polymorphisms in DNA repair pathways [20].

Commonly detected OH-PAHs include urinary metabolites of naphthalene, fluorene, phenanthrene, and pyrene. Importantly, naphthalene, fluorene, and phenanthrene are low-molecular weight PAHs with 2–3 aromatic rings, while pyrene is considered high-molecular weight with 4 aromatic rings [1, 21]. Evidence suggests that higher-molecular weight PAHs with ≥4 aromatic rings, such as benzo[a]pyrene, are primarily excreted through feces and are typically not detectable in urinary samples [22–25]. 1-hydroxypyrene (1-PYR) has been widely

used as a proxy for total PAHs exposure given its moderate-high correlation with both low and high-molecular weight PAHs [24, 26, 27]. However, 1-PYR alone may not be an adequate proxy given that the magnitude and type of PAHs vary by PAH sources. For instance, 1-PYR levels are strongly linked to dietary patterns, whereas naphthalene and fluorene levels are strongly linked to tobacco and wood smoke exposure [28–31]. For this reason, several studies have highlighted the need for assessment of multiple OH-PAHs to obtain a more comprehensive representation of PAHs body burden across multiple exposure sources [30, 32].

Urinary metabolites of naphthalene, fluorene, phenanthrene, and pyrene have been highly detected in the U.S. population across two decades of the National Health and Nutrition Examination Survey (NHANES) biomonitoring [24]. Short half-lives coupled with high levels of detection indicate likely chronic exposure from multiple PAHs sources in the U.S. population. However, despite evidence for chronic exposure and prior evidence suggesting a link with chronic and often fatal disease, the impact of PAHs on mortality in the general population has received limited attention. To date, only one study has examined this relationship using a limited sample of the NHANES population [33]. This study did not account for urinary dilution of PAH concentrations and the number of mortality endpoints was sparse, resulting in highly imprecise estimates for several PAHs. Furthermore, Chen et al. did not account for the dense correlation structure between PAH metabolites.

In this study, we aimed to prospectively evaluate associations between a sum of multiple, commonly detected urinary OH metabolites, as a proxy for total PAH exposure from various sources, and mortality endpoints in the U.S. adult population, with consideration of differences by age, gender, race/ethnicity, smoking and socioeconomic status. Additionally, as a secondary aim, we leverage quantile g-computation, an analytic approach to evaluate the risk associated with environmental mixtures, to gain insight into the joint association of OH metabolites of commonly detected PAHs, individual OH-PAH contributions, and potential heterogeneity.

## Methods

### Study population and data collection

NHANES is conducted by the National Center for Health Statistics (NCHS) within the U.S. Centers for Disease Control and Prevention (CDC). Individuals are selected using a complex, multistep, probability-based sampling design and are considered representative of the civilian, non-institutionalized U.S. population. NHANES oversamples certain population sub-groups including African Americans, Mexican Americans, low-income white Americans, and 12-19-year-olds. Since 1999, the survey has been continuous, and each cycle covers two years. The survey comprises a household interview and a standardized physical examination in a mobile examination center. Household interviews collect detailed demographic, socioeconomic, dietary, and health-related information. Physical examinations comprise of physiological measurements, medical and dental examinations, biological sampling, and laboratory tests. We included participants from seven consecutive survey cycles (2001–2014), totaling 72,126 individuals. NCHS' Institutional Review Board approved each NHANES cycle, and survey participants provided informed consent.

NHANES analyzes blood or urine environmental chemical data on a one-third random sub-sample of survey participants. There were 12,316 individuals at least 20 years old with spot urine sample OH-PAHs biomarker data across the 2001–2014 survey cycles.

### Exclusion criteria

We excluded 6,763 individuals less than 20 years old at the time of interview or physical examination because mortality ascertainment data is not publicly-available for individuals less than

18 years of age, and children and adolescents differ from adults in physiological profile, which may influence exposure dynamics and response (S1 Fig). We subsequently excluded individuals (N = 21) missing information on vital status as ascertained through the National Death Index (NDI) and individuals (N = 982) missing data on any of the eight OH-PAHs or urinary creatinine. Lastly, we excluded individuals missing information on any covariates defined below under statistical analysis (N = 1,539), improbable follow-up time (N = 6) and accidental deaths (N = 29). Our final analytical sample for all causes of death was 9,739 participants. To assess for potential for selection bias due to exclusion criteria, we compared distribution of exposure and covariates in our study population by mortality status before and after application of exclusion criteria (S1 Table).

## Exposure assessment

A single spot urine sample was collected from participants, stored at -20˚C, and sent for analysis to the National Center for Environmental Health at CDC. Eight OH-PAHs from four parent PAHs (naphthalene, fluorene, phenanthrene, and pyrene) were measured across all seven survey cycles: 1-hydroxynaphthalene/naphthol (1-NAP); 2-hydroxynaphthtalene/naphthol (2-NAP); 2-hydroxyfluorene (2-FLUO); 3-hydroxyfluorene (3-FLUO); 1-hydroxyphenanthrene (1-PHEN); 2-hydroxyphenanthrene (2-PHEN); 3-hydroxyphenanthrene (3-PHEN); and 1-PYR.

OH-PAHs were measured using enzymatic deconjugation followed by automated solid phase extraction. Quantification of analytes was done through capillary isotope dilution gas chromatography (GC) and high-resolution mass spectroscopy (HRMS) for cycles 2001–2008 [34, 35], GC tandem mass spectrometry (GC-MS/MS) for cycles 2009–2012 [36], and online solid-phase extraction and high-performance liquid chromatography tandem MS (online SPE-HPLC-MS/MS) for 2013–2014 [37].

For the 2013–2014 cycle, online SPE-HPLC-MS/MS quantitated 2-PHEN and 3-PHEN together, which is equivalent to the sum of the two urinary metabolites from previous cycles. We present the two metabolites as $\Sigma$ [2-PHEN & 3-PHEN].

For each OH-PAH, concentrations (ng/L) below the highest methodological LOD across survey cycles were assigned the value of that highest LOD divided by square root of two. To standardize across differential molecular weights of OH-PAHs, concentrations (ng/L) were converted to molar concentrations (nmol/L). All OH-PAHs were log-transformed following visual and quantitative ascertainment of right skewness. Upon log transformation, all OH-PAHs were normally distributed.

For assessment of mortality in relation to PAHs exposure, a $\Sigma$ OH-PAHs (nmol/L) was created. This was calculated as the molar sum of 1-NAP, 2-NAP, 2-FLUO, 3-FLUO, 1-PHEN, 2&3 PHEN, and 1 PYR. $\Sigma$ OH-PAHs was subsequently log-transformed.

## Mortality outcomes

Mortality status for the study population was ascertained through the U.S. National Death Index (NDI), a compiled index of death record information from local and state vital statistics offices. NCHS periodically links NHANES surveys with death certificate records from the NDI. We utilized the publicly available Linked Mortality Files provided by NCHS, updated through December 31, 2015, which served as end of follow-up (i.e., point of administrative censoring) (https://www.cdc.gov/nchs/data-linkage/mortality-public.htm). All NDI data is based on death certificates coded to International Classification of Diseases version 10 codes (ICD-10).

We examined three main outcomes: all-cause mortality, cancer-specific mortality (ICD-10: C00-C97), and cardiovascular-specific mortality (ICD-10: I00-I09, I11, I13, I20-I51). Our cancer-specific mortality outcome aimed to identify deaths from incident cancer after baseline. Therefore, we excluded 877 additional individuals with self-reported history of cancer at baseline, resulting in an analytic sample of N = 8,862 for the cancer-specific mortality analyses. Analogously, for our cardiovascular mortality outcome, we excluded 764 individuals with self-reported history of coronary heart disease, heart attack, and congestive heart failure at baseline, resulting in an analytic sample of N = 8,975 for the cardiovascular-specific mortality analyses.

## Statistical analysis

In descriptive analyses, we computed geometric means (GM) and standard errors of the geometric mean (GSE) for Σ H-PAHs (nmol/L) by mortality status and relevant covariates. Frequencies of categorical covariates and GM of continuous covariates were additionally computed by mortality status. To account for NHANES complex, multistage, probability sampling design in our analyses, we used PAH subsample weights, stratification variables, and primary sampling units provided by NCHS. A 14-year modified sample weight across seven NHANES cycles was created for each participant based on NCHS guidance (https://www.cdc.gov/nchs/tutorials/environmental/critical_issues/limitations/Info3.htm). We produced Pearson correlation coefficients for all OH-PAHs.

The minimally sufficient adjustment set (covariates) was determined *a priori* through a directed acyclic graph approach [38] (**S2 Fig**), including age at time of interview (years), sex (male, female), race/ethnicity (non-Hispanic White, non-Hispanic Black, Hispanic, other), smoking status (current active smoker or environmental tobacco smoke [ETS] exposed, not active smoker and ETS unexposed), urinary creatinine (g/L), body mass index (BMI) (underweight, normal weight, overweight, and obese according to WHO guidelines (($kg/m^2$); [39]), educational attainment (less than high school, high school graduate, some college or above), family income to poverty ratio (income greater than poverty level, income less than or equal to poverty level), and survey cycle years (cycles 1–7). Our primary method to account for urine dilution was to add urinary creatinine (g/L) as a covariate to all models, as recommended by Barr et al. [40]. Urinary creatinine was measured in urine spot samples through automated colorimetric determination on a Beckman Synchron CX3 clinical analyzer. Smoking status at baseline was derived from a combination of self-reported household interview questions about current smoking status, as well as a laboratory measurement of serum cotinine, indicating exposure to ETS. Serum cotinine (ng/mL) was measured using an isotope dilution HPLC coupled with atmospheric pressure chemical ionization tandem mass spectrometry. Individuals were categorized as current active smokers or ETS exposed based on self-report (Q1: "Smoked at least 100 cigarettes in life?"; Q2: "Age started smoking cigarettes regularly"; Q3: "Do you now smoke cigarettes?") or serum cotinine concentrations >10 ng/mL.

For analyses of Σ OH-PAHs and mortality endpoints, we used multivariable survey-weighted Cox proportional hazards regression. Adjusted hazard ratios ($HR_{adj}$) and 95% confidence intervals (CI) are reported. Our models use age, in years, as the timescale for analyses (age at NHANES interview was age of entry into study and age at mortality or age at administrative censoring was the age of exit from study). To verify that the proportional hazards assumption was met, we ran diagnostic tests, including an assessment of interaction between exposure and time in addition to visual inspection of degree of parallelism plotting log cumulative hazard versus log time. Σ OH-PAHs was modeled as a continuous variable in main and subgroup analyses and additionally modeled using a quartile categorization in main analyses. Linear trend was tested by assigning median value of Σ OH-PAH quartiles to the respective

quartiles followed by a multivariable model incorporating assigned median values as a continuous exposure variable.

Subgroup analyses were performed for age (<60 years, ≥60 years), gender (male, female), smoking status (current active smoker or ETS exposed, not current active smoker or ETS exposed), family income to poverty ratio (family income at or below poverty level, family income above poverty level) and race/ethnicity (non-Hispanic White, non-Hispanic Black, Hispanic, other). Effect modification was assessed through interaction term(s) between Σ OH-PAHs and sub-group levels followed by assessment of interaction term p-values (two-level sub-group) or the group F-statistic for unequal slopes (multiple-level sub-group). A p-value of less than 0.05 was considered a statistically significant interaction, though we additionally considered differences in magnitude of association across strata as well as biological plausibility.

In sensitivity analyses, we excluded individuals who died within the first and second year of follow-up to examine for the possibility of mortality within these years driving effects. We additionally assessed the impact of urine dilution using both urinary creatinine-corrected OH-PAHs concentrations, as well as adjusting for urinary creatinine as a covariate, as recommended by Weinberg et al. [41]. Lastly, we conducted analyses including prevalent cases of CVD and cancer (i.e. self-reported history of CVD or cancer at baseline) to assess the impact exclusion of such cases may have had on our risk estimates.

### Mixtures analysis

For analyses of the overall joint effect of the eight OH-PAHs and contributions of individual OH-PAHs, we used Cox proportional hazards quantile g-computation [42]. Quantile g-computation estimates the change in mortality risk for a one quantile simultaneous increase in the eight OH-PAHs. Importantly, quantile g-computation allows flexibility in direction and magnitude of effect across OH-PAHs, while controlling for mutual confounding among the OH-PAHs. In addition to the overall joint effect, quantile g-computation generates weights for each constituent of the mixture. Weights for OH-PAHs represent the proportion of the total effect for each OH-PAH when all OH-PAHs have an effect in the same direction or the proportion of the positive or negative partial effect when coefficients are in different directions. We chose a quartile categorization of exposure for internal consistency with other presented models. We were not able to survey-weight the quantile g-computation models because specification of sampling strata is a feature not yet developed for this method and NHANES weights are valid only when sampling cluster and strata are defined. We also computed multi-pollutant (mutually-adjusted) survey weighted Cox proportional hazard models for the eight OH-PAHs for each mortality endpoint for comparison to quantile g-computation results. Quantile g-computation and multi-pollutant models all included adjustment for covariates used in our main models.

All analyses were conducted in Statistical Analysis Software version 9.4 (Cary, North Carolina) and R software version 3.6.2 (cran.r-project.org).

## Results

Of the 9,739 participants included in the analysis, 934 participants (9.60%) died (**Table 1**). Follow-up time among the deceased was shorter than those alive at time of censoring (5.63 and 7.56 years, respectively). Across the seven NHANES cycles, OH-PAHs were detected in nearly all participants (detection frequency (DF) > 98%), except for 1-PYR (DF = 60.95%). ΣOH--PAHs were higher among deceased participants compared to alive participants. Compared to living participants, those deceased participants were more likely to be older, have a lower

**Table 1. Baseline descriptive characteristics in NHANES 2001–2014 (N = 9,739) by mortality status.**

| | | | Alive (N = 8,805) | Deceased (N = 934) |
|---|---|---|---|---|
| | | >LOD (%) | GM (GSE) | |
| Urinary OH-PAH (ng/L) | | | | |
| | 1-naphthalene | 99.85 | 2287.34 (61.24) | 3560.92 (258.57) |
| | 2-naphthalene | 99.97 | 3759.81 (87.54) | 3126.28 (172.27) |
| | 3-hydroxyfluorene | 98.44 | 112.68 (2.92) | 110.58 (7.72) |
| | 2-hydroxyfluorene | 99.97 | 277.46 (6.64) | 298.24 (18.11) |
| | 1-hydroxyphenanthrene | 99.67 | 135.78 (2.24) | 136.75 (5.55) |
| | Σ 2- & 3-hydroxyphenanthrene | 99.82 | 151.00 (2.61) | 163.57 (8.20) |
| | 1-hydroxypyrene | 60.95 | 115.05 (1.75) | 89.69 (3.13) |
| | Σ OH-PAHs[a] | --- | 8192.57 (187.10) | 9134.58 (534.17) |
| Age at baseline (years) | | | 45.03 (0.25) | 66.01 (0.65) |
| Follow up time (years) | | | 7.56 | 5.63 |
| Urinary creatinine (g/L) | | | 1.22 (0.01) | 1.12 (0.03) |
| | | | Frequency (SE) | |
| Gender | | | | |
| | Male | | 48.49 (0.58) | 54.40 (2.11) |
| | Female | | 51.51 (0.58) | 45.60 (2.11) |
| Race/ethnicity | | | | |
| | Non-Hispanic white | | 69.74 (1.33) | 77.61 (2.00) |
| | Non-Hispanic black | | 11.02 (0.70) | 10.52 (1.04) |
| | Hispanic | | 12.97 (0.98) | 9.23 (1.47) |
| | Other | | 6.28 (0.42) | 2.64 (0.61) |
| Educational attainment | | | | |
| | Less than high school graduate | | 16.44 (0.72) | 30.19 (1.78) |
| | High school graduate | | 22.82 (0.66) | 29.57 (1.36) |
| | Some college or above | | 60.74 (1.03) | 40.24 (1.79) |
| Smoking status[b] | | | | |
| | Not active smoker or ETS exposed | | 71.79 (0.72) | 72.27 (1.72) |
| | Active smoker or ETS exposed | | 28.21 (0.72) | 27.74 (1.72) |
| Body mass index (kg/m$^2$) | | | | |
| | Underweight (<18.5) | | 1.55 (0.17) | 2.67 (0.78) |
| | Normal (18.5–24.9) | | 30.59 (0.69) | 29.53 (1.73) |
| | Overweight (25–29.9) | | 33.36 (0.78) | 33.32 (1.64) |
| | Obese (≥30) | | 34.50 (0.67) | 34.49 (1.90) |
| Family poverty status[c] | | | | |
| | Above the poverty threshold | | 86.15 (0.57) | 82.17 (1.45) |
| | At or below the poverty threshold | | 13.85 (0.57) | 17.83 (1.45) |

Abbreviations: ETS = environmental tobacco smoke; GM = geometric mean; GSE = standard error of the GM; LOD = limit of detection; SE = standard error.

[a]Σ OH-PAHs include all eight urinary hydroxylated PAH metabolites from four parent compounds (naphthalene, fluorene, phenanthrene, pyrene)

[b]Smoking status defined as current active smoker based on questionnaire, or serum cotinine concentrations >10 ng/mL, defined at ETS exposure

[c]Poverty status is calculated as the ratio of the family's self-reported income to the family's poverty threshold

educational attainment, be non-Hispanic white and be at or below the poverty level. Across survey cycles (2001–2014), PAHs exposure was highest in 2005–2006, although no clear trends were apparent. We did not find any differences in baseline characteristics of the study population prior to and after application of exclusion criteria (S1 Table)."

OH-PAHs were moderately to highly correlated (r = 0.51–0.95), both within and across a OH-PAH's parent compound (S2 Table). Participants with higher GM ΣOH-PAH concentrations were more likely to be <60 years old, male, current active smoker or ETS exposed, and underweight or obese. Higher GM ΣOH-PAH participants were also more likely to be non-Hispanic black, have lower educational attainment, and be living at or below the poverty threshold (Table 2). A $\log_{10}$ increase in ΣOH-PAHs was positively associated with increased

**Table 2. Baseline Σ OH-PAHs (ng/L)[a] by participant characteristics (N = 9,739).**

| Participant characteristics | | GM (GSE) of Σ PAHs (ng/L) |
|---|---|---|
| Age at baseline | | |
| | 20 to <60 years | 8808.11 (219.68) |
| | ≥60 years | 6621.75 (216.29) |
| Race/ethnicity | | |
| | Non-Hispanic white | 7819.83 (239.93) |
| | Non-Hispanic black | 12305.00 (374.41) |
| | Hispanic | 8415.46 (273.22) |
| | Other | 7136.77 (463.27) |
| Gender | | |
| | Male | 9030.31 (229.93) |
| | Female | 7568.09 (224.39) |
| Educational attainment | | |
| | <High school graduate | 11040.00 (448.91) |
| | High school graduate | 10168.00 (383.73) |
| | Some college or above | 6983.47 (193.79) |
| Smoking status[b] | | |
| | Not active smoker or ETS exposed | 5656.07 (106.50) |
| | Active smoker or ETS exposed | 21601.00 (710.26) |
| Body mass index (kg/m$^2$) | | |
| | Underweight (<18.5) | 9516.43 (1418.73) |
| | Normal (18.5–24.9) | 7484.54 (298.65) |
| | Overweight (25–29.9) | 7792.44 (232.88) |
| | Obese (≥30) | 9440.58 (281.32) |
| Family poverty status[c] | | |
| | Above the poverty threshold | 7773.32 (187.41) |
| | At or below the poverty threshold | 11858.00 (483.10) |
| NHANES survey cycle | | |
| | 2001–2002 | 6748.33 (511.10) |
| | 2003–2004 | 8615.59 (594.79) |
| | 2005–2006 | 10492.00 (622.53) |
| | 2007–2008 | 8862.80 (672.71) |
| | 2009–2010 | 7952.30 (392.26) |
| | 2011–2012 | 7950.09 (390.85) |
| | 2013–2014 | 7819.23 (301.47) |

Abbreviations: ETS = environmental tobacco smoke; GM = geometric mean; GSE = standard error of the GM; NHANES = National Health and Nutrition Examination Survey

[a]Σ OH-PAHs include 8 urinary PAH metabolites from four parent compounds (naphthalene, fluorene, phenanthrene, pyrene)

[b]Smoking status defined as current active smoker based on questionnaire, or serum cotinine concentrations >10 ng/mL.

[c]Poverty status is calculated as the ratio of the family's self-reported income to the family's poverty threshold

**Table 3. Association between log-transformed ΣOH-PAHs (nmol/L) and all-cause, cancer-specific, and CVD-specific mortality, adjusted for covariates[a].**

| Mortality type | | Continuous (per $\log_{10}$-increase) | Quartile 1 | Quartile 2 | Quartile 3 | Quartile 4 | p-trend[b] |
|---|---|---|---|---|---|---|---|
| All causes (N = 9739) | Cases | 934 | 240 | 233 | 195 | 266 | |
| | ΣOH-PAHs (nmol/L), median | 1.72 | 1.17 | 1.56 | 1.89 | 2.40 | |
| | $HR_{adj}$ (95% CI) | 1.39 (1.21, 1.61) | Ref. | 1.16 (0.89, 1.51) | 1.17 (0.89, 1.54) | 1.66 (1.32, 2.09) | <0.001 |
| Cancer-specific (N = 8862) | Cases | 159 | 34 | 41 | 37 | 47 | |
| | ΣOH-PAHs (nmol/L), median | 1.72 | 1.18 | 1.56 | 1.90 | 2.40 | |
| | $HR_{adj}$ (95% CI) | 1.15 (0.79, 1.69) | Ref. | 1.14 (0.66, 1.96) | 1.35 (0.74, 2.45) | 1.56 (0.80, 3.04) | 0.17 |
| CVD-specific (N = 8975) | Cases | 108 | 32 | 25 | 22 | 29 | |
| | ΣOH-PAHs (nmol/L), median | 1.72 | 1.17 | 1.56 | 1.89 | 2.39 | |
| | $HR_{adj}$ (95% CI) | 1.49 (0.94, 2.33) | Ref. | 1.36 (0.55, 3.34) | 1.11 (0.47, 2.59) | 1.79 (0.68, 4.71) | 0.28 |

Abbreviations: CVD = cardiovascular disease

[a]Models adjusted for age (years), gender (male/female), race/ethnicity (non-Hispanic white, non-Hispanic black, Hispanic, other race/ethnicity), smoking status (current, not-current), BMI ($kg/m^2$), survey cycle (cycles 1–7), educational attainment (<high school, high school graduate, some college or above), family poverty status (above, at or below family poverty threshold), and urinary creatinine (g/L)

[b]Computed using a 'continuous' exposure created out of medians of each quartile of $\log_{10}$ ΣOH-PAHs

all-cause mortality ($HR_{adj}$: 1.39 [95%CI: 1.21, 1.61]; N = 934 deaths) (**Table 3**). In categorical analysis comparing the highest and lowest quartiles of $\log_{10}$ ΣOH-PAHs, all-cause mortality was elevated ($HR_{adj}$ for Q4 vs. Q1: 1.66 [95%CI: 1.32, 2.09]), with a positive exposure-response relationship ($P_{trend}$ < 0.001). An elevated but non-significant association was seen for ΣOH-PAHs and cancer-specific mortality when modeled continuously ($HR_{adj}$: 1.15 [95%CI: 0.79, 1.69]; N = 159 deaths) and categorically ($HR_{adj}$ for Q4 vs. Q1: 1.56 [95%CI: 0.80, 3.04]). $HR_{adj}$ for cardiovascular-specific mortality (N = 108 deaths) were 1.49 (95%CI: 0.94, 2.33) and 1.79 (95%CI: 0.68, 4.71) when modeled continuously and as Q4 vs. Q1 exposure categories, respectively. While no significant exposure-response relationship was seen for cancer-specific or cardiovascular-specific mortality, there was evidence for a monotonic, positive exposure-response relationship for cancer-specific mortality, albeit limited in precision.

In subgroup analyses (**Table 4**), substantial differences were observed by smoking ($P_{interaction}$ = 0.02) and age ($P_{interaction}$ = 0.06) for all-cause mortality, by gender ($P_{interaction}$ = 0.05) for cancer-specific mortality, and by race/ethnicity for cardiovascular-specific mortality ($P_{interaction}$ = 0.02). Mortality across all three endpoints was higher among active smokers and ETS exposed, participants younger than 60 years, and non-Hispanic blacks relative to non-active or passive smokers, participants at or above 60 years, and non-Hispanic white and other race/ethnicities, respectively. Women had an increased risk of cancer-specific mortality ($HR_{adj}$ = 1.53 [95%CI: 1.05, 2.23]) compared to men ($HR_{adj}$ = 0.80 [95%CI: 0.45, 1.42]). Participants living at or below the poverty level had an increased risk of cardiovascular-specific mortality ($HR_{adj}$ = 2.83 [95%CI: 1.14, 6.99]) compared to those living above the poverty level ($HR_{adj}$ = 1.28 [95%CI: 0.78, 2.11]).

In models excluding deaths occurring within 1- or 2-years of baseline (**S3 Table**), we observed similar estimates for each outcome of interest, except that risk of cardiovascular-specific mortality was elevated compared to models used in main analyses (**Table 3**). When ΣOH-PAHs were both creatinine-corrected (nmol/g Cre) and creatinine included as a covariate in models, results for our three mortality outcomes were generally similar (**S4 Table**).

**Table 4. Assessment of effect heterogeneity or effect measure modification for log-transformed ΣOH-PAHs (nmol/L) concentrations and mortality by age, gender, family poverty status, smoking status, and race/ethnicity[a].**

| | All-cause mortality | | | Cancer-specific mortality | | | CVD-specific mortality | | |
|---|---|---|---|---|---|---|---|---|---|
| | events/N | HR$_{adj}$ (95%CI) | p$^{b}$ | events/N | HR$_{adj}$ (95%CI) | p$^{b}$ | events/N | HR$_{adj}$ (95%CI) | p$^{b}$ |
| **Age at baseline** | | | | | | | | | |
| <60 years | 211/6709 | 1.59 (1.31, 1.93) | | 59/6423 | 1.35 (0.88, 2.07) | | 22/6507 | 1.60 (0.99, 2.60) | |
| ≥60 years | 723/3030 | 1.32 (1.13, 1.54) | 0.06 | 100/2439 | 1.04 (0.69, 1.56) | 0.12 | 86/2468 | 1.44 (0.88, 2.37) | 0.62 |
| **Gender** | | | | | | | | | |
| Male | 554/4818 | 1.37 (1.12, 1.67) | | 99/4409 | 0.80 (0.45, 1.42) | | 61/4337 | 1.46 (0.82, 2.61) | |
| Female | 380/4921 | 1.41 (1.18, 1.69) | 0.80 | 60/4453 | 1.53 (1.05, 2.23) | 0.05 | 47/4638 | 1.50 (0.84, 2.67) | 0.95 |
| **Family poverty status** | | | | | | | | | |
| >Federal poverty level | 739/7709 | 1.36 (1.17, 1.58) | | 117/6955 | 1.20 (0.78, 1.83) | | 82/7129 | 1.28 (0.78, 2.11) | |
| ≤Federal poverty level | 195/2030 | 1.56 (1.15, 2.11) | 0.40 | 42/1907 | 0.99 (0.50, 1.98) | 0.63 | 26/1846 | 2.83 (1.14, 6.99) | 0.11 |
| **Smoking status** | | | | | | | | | |
| Nonsmoker/no ETS exposure | 683/7068 | 1.27 (1.08, 1.50) | | 110/6369 | 1.02 (0.61, 1.72) | | 77/6501 | 1.28 (0.76, 2.15) | |
| Active smoker/ETS exposure | 251/2671 | 1.96 (1.43, 2.67) | 0.02 | 49/2493 | 1.59 (0.86, 2.94) | 0.29 | 31/2474 | 2.41 (1.09, 5.32) | 0.14 |
| **Race/ethnicity** | | | | | | | | | |
| Non-Hispanic white | 572/4703 | 1.39 (1.18, 1.63) | | 75/4095 | 1.10 (0.70, 1.74) | | 61/4225 | 1.58 (0.99, 2.50) | |
| Non-Hispanic black | 170/2006 | 1.57 (1.08, 2.29) | | 37/1865 | 1.30 (0.63, 2.68) | | 25/1865 | 1.96 (0.79, 4.88) | |
| Other | 192/3030 | 1.23 (0.89, 1.69) | 0.58 | 47/2902 | 1.23 (0.76, 2.01) | 0.88 | 22/2895 | 0.48 (0.18, 1.28) | 0.02 |

Abbreviations: CVD = cardiovascular disease; ETS = environmental tobacco smoke

[a]All models adjusted for BMI (kg/m$^2$), NHANES survey cycle (cycles 1–7), educational attainment (<high school, high school graduate, some college or above), and urinary creatinine (g/L). Age-stratified models additionally adjusted for age (years), gender (male/female), race/ethnicity (non-Hispanic white, non-Hispanic black, Hispanic, other race/ethnicity), smoking status (current, not-current), and family poverty status (above, at or below family poverty threshold). Gender-stratified models additionally adjusted for age, race/ethnicity, smoking status, and family poverty status; smoking-stratified models additionally adjusted for age, race/ethnicity, gender, and family poverty status; family poverty status additionally adjusted for age, race/ethnicity, gender, and smoking status; race/ethnicity-stratified models additionally adjusted for age, gender, smoking status, and family poverty status; for race/ethnicity stratified models, Hispanic and other race/ethnicities were collapsed into "other" for purposes of heterogeneity analyses due to event sample size limitations.

[b]Test for heterogeneity was computed using either the p-value for the product term in the model (age, gender, smoking, poverty models) or through the F-test (race/ethnicity model)

When including baseline cases, the adjusted risk estimates slightly increased for cancer-specific mortality, and decreased for CVD-specific mortality, but remained non-significant and elevated (**S5 Table**).

In our quantile g-computation analysis (**Table 5**), HR$_{adj}$ for the overall joint effect (simultaneous quartile increase) of the eight OH-PAHs were 1.15 (95%CI: 1.02, 1.31), 1.41 (95%CI:

**Table 5. Joint effect of OH-PAHs[a] on mortality outcomes[b] using quantile g-computation.**

| Mortality type | OH-PAHs categorization | HR$_{adj}$ (95% CI) | p-value |
|---|---|---|---|
| All-cause | Quartile | 1.15 (1.02, 1.31) | 0.03 |
| Cancer-specific | Quartile | 1.41 (1.05, 1.90) | 0.02 |
| CVD-specific | Quartile | 0.98 (0.66, 1.47) | 0.93 |

[a] OH-PAHs were 1-hydroxynaphthalene, 2-hydroxynaphthalene, 2-hydroxyfluorene, 3-hydroxyfluorene, 1-hydroxyphenanthrene, 2&3-hydroxyphenanthrene, 1-hydroxypyrene

[b]Models adjusted for age (years), gender (male/female), race/ethnicity (non-Hispanic white, non-Hispanic black, Hispanic, other race/ethnicity), smoking status (current, not-current), BMI (underweight, normal, overweight, obese), survey cycle (cycles 1–7), educational attainment (<high school, high school graduate, some college or above), family poverty status (above, at or below family poverty threshold), and urinary creatinine (g/L)

1.05, 1.90), and 0.98 (95% CI: 0.66, 1.47) for all-cause, cancer-specific, and cardiovascular-specific mortality, respectively. In quantile g-computation, 1-NAP, 2-NAP, 2-FLUO, 2&3-PHEN, and 1-PYR were positively weighted while 3-FLUO and 1-PHEN were negatively weighted for all-cause mortality. All OH-PAHs but PHEN were positively weighted for cancer-specific mortality, while only 2&3-PHEN, 1-NAP, and 2-FLUO were positively weighted for cardiovascular-specific mortality (**S3 Fig**). There was some concordance between direction of weights from quantile g-computation and direction of effects in multi-pollutant models across endpoints (**S4 Table**).

## Discussion

In a large representative sample of the U.S. adult population, our study found that an increase in ΣOH-PAHs urinary concentrations was associated with higher all-cause mortality. For cause-specific deaths, ΣOH-PAHs were associated with non-significant increases in cardiovascular-specific mortality, and to a lesser extent, cancer-specific mortality. We observed substantial differences in PAH deleterious effects by age and smoking status for all-cause mortality, gender for cancer-specific mortality, and race/ethnicity for cardiovascular-specific mortality. Our results also identified disparities in risk of cardiovascular-specific mortality for participants who were non-Hispanic black and those living at or below the poverty level. When examining the joint effect using quantile g-computation, a quartile simultaneous increase of all eight OH-PAHs was associated with increased all-cause and cancer-specific mortality; null effects were seen for cardiovascular-specific mortality. There was consistency in the direction and significance of association for all-cause mortality across our quantile g-computation and sum OH-PAHs approaches. In quantile g-computation, we observed mostly positive and some negative contributions of OH-PAHs for endpoints.

Our findings for all-cause mortality and cardiovascular-specific mortality support an emerging body of epidemiologic literature on the health impact of PAHs, including Chen et al. (although we note that the studies are not directly comparable due to lack of urinary dilution, differences in study population size, and analytic approach). All-cause mortality represents a broad range of disease categories. Though the apportionment of specific causes to overall mortality was not examined as such data is not publicly available, leading causes of death (excluding unintentional injury) in the U.S. include lower respiratory disease, cerebrovascular disease, cardiovascular disease, and cancer [43]. In occupational settings, PAHs have been linked to fatal ischemic heart disease and fatal respiratory disease [4, 5]; however, these PAHs are often found at higher concentrations compared to the general population. Evidence of PAH exposure and cardiovascular-specific mortality in non-occupational populations is not well-characterized. A Chinese population-based cohort reported an association between OH-PAHs and 10-year atherosclerotic cardiovascular disease risk score, but not coronary heart disease [44]. Previous NHANES studies have reported positive associations between PHEN and prevalence of self-reported cardiovascular disease [45] as well as positive associations between OH-PAHs and prevalence of self-reported CVD [46]. Similarly, we found no mortality studies examining fatal respiratory diseases and PAH biomarkers, though cross-sectional studies have reported significant decreases in respiratory measures associated with commonly measured OH-PAHs [9, 47]. Although understanding of mechanisms through which PAHs impact human health remains a work in progress, one important mechanism is thought to be PAH induced aryl hydrocarbon receptor (AhR) activation and ensuing inflammatory, oxidative, and genotoxic effects [48–50]. AhR is highly expressed in liver, adipose, and bronchial tissue and is pivotal in cardiac development and function [51–54].

PAH-related health disparities by race/ethnicity (non-Hispanic black) and poverty status (at or below the poverty level) were seen for cardiovascular-related mortality. In the US, non-Hispanic black and individuals below the poverty level have higher cardiovascular mortality compared to other race/ethnicities and individuals above the poverty level [55]. We also saw overall higher OH-PAH concentrations among these minority population groups, indicating higher overall PAH exposure. Trend analysis from non-smoking NHANES participants from 2001–2014 also revealed increasing concentrations of PAHs among minority groups, including Non-Hispanic blacks and Mexican Americans [56]. One possible explanation for the observed disparities for PAH exposure and cardiovascular mortality is the interplay of psychosocial stressors (e.g., allostatic load) on cardiovascular disease among these population groups [57]. PAHs may be linked to some allostatic biomarkers; an NHANES study found an association with C-reactive protein and urinary PAH biomarkers [46]. Findings by smoking status in relation to all-cause mortality highlighted that PAH-related mortality risk from non-tobacco smoke sources is substantial and that PAHs from tobacco smoke further elevate mortality risk, possibly through differential PAHs profile and/or synergy among PAHs and other deleterious components of tobacco smoke such as carbon monoxide and formaldehyde.

We observed weak elevated risk of cancer-specific mortality with increasing OH-PAH concentrations, with substantial differences where there was elevated risk among women and slightly reduced risk among men. Cancer is many diseases with different etiologies. Though tumor sites were not accessible in the publicly available NDI dataset, leading causes of mortality by cancer subtype in the US, including lung, breast, and colorectal cancer, have relevance to PAH exposure. Urinary biomarkers of PAH exposure (2-NAP, 1-PYR) have been associated with increased lung cancer, especially in the presence of biomarkers of oxidative stress [58]. For breast cancer, several case-control investigations have reported positive associations with breast cancer incidence [6, 7, 59], although Lee et al. reported slight negative associations [60]. As a check on the possibility of undetected, potentially advanced cancer driving effect patterns, we excluded cancer deaths within 1 and 2 years of cohort entry in sensitivity analyses and found that doing so had no influence on findings. Our pattern of findings supports some of the aforementioned literature, although it should be noted that these studies [6, 7, 59] reported on breast cancer incidence, while we report on cancer mortality from incident cancer. Even if breast cancer is a major contributor to cancer mortality in this representative sample of the US population, the effect of PAHs on breast cancer incidence and mortality may not be directly comparable. With regards to gender-specific differences in cancer outcomes in relation to PAH exposure in this study as well as in Chen et al. [33], Guo et al. [61] suggested that women are more susceptible than men to chromosomal damage and oxidative stress consequences of PAH (Σ [NAP, FLUO, PHEN, PYR]) exposure [61].

The lack of consistency in some results between ΣOH-PAHs and our mixtures analysis using quantile g-computation highlight differences in approaches reflective of their divergent purposes. ΣOH-PAHs approximate the total body burden of commonly detected OH-PAHs without further resolution by PAHs mixture composition. Environmental PAHs co-occur as complex mixtures of tens of PAHs, many of which vary in relative toxicity and are poorly detected in biological samples. Therefore, our ΣOH-PAH measure serves as a crude proxy for total PAH exposure, irrespective of individual contributions. By comparison, quantile g-computation in this study estimates the impact of simultaneous quantile increase of all measured OH-PAHs, and examines individual contributions of each metabolite to that joint effect. Quantile g-computation weights, a measure of individual metabolite contribution, are robust to collinearity among measured OH-PAHs where traditional multipollutant regression models tend to perform poorly [62]. From the parent set of examined OH-PAHs, naphthalene is classified by the NTP [16] as reasonably anticipated to be a human carcinogen and by IARC as

probably carcinogenic to humans (Group 2B) while fluorene, phenanthrene, and pyrene are deemed non-classifiable as to their carcinogenicity in humans (Group 3) [15]. While multipollutant models showed conflicting negative and positive associations for 1-NAP and 2-NAP, respectively, quantile g-computation positive weights for both are consistent with IARC classification. Additionally, 2-FLUO, 3-FLUO, and 1-PYR were found to contribute positive weights for cancer mortality while 1-PHEN and 2&3-PHEN were found to contribute negative weights cancer mortality. Although Chen et al. previously reported associations for mortality endpoints by individual OH-PAHs that are considered in this study, we note that they don't correct for mutual confounding among measured OH-PAHs (**S2 Table**); moreover, previously reported effect estimates are highly imprecise [33]. Given the paucity of human evidence on individual OH-PAHs effects, weights from our quantile g-computation results are intended to inform future analyses, with the aforementioned limitations and also the possibility of confounding by unmeasured factors, including unmeasured PAHs, in mind given that PAHs are a highly complex mixture.

Our study had several limitations. The use of a one-time spot urinary sample may not reflect PAH exposure over longer time frames. Biomonitoring studies demonstrate moderate temporal reliability of spot urinary measurements for PAHs; Dobraca et al. [63] reported intraclass correlation coefficients (ICC) ranging from 0.07 to 0.53 across PAH metabolites considered in this study and ICCs for most PAH metabolites ranged between 0.35 and 0.50 across multiple years. Importantly, concentration from the single spot sample reliably ranked exposure into quartiles consistent with study's multi-year average [63]. Li et al. [18] reported an ICC of 0.55 for 1-PYR. Reliability was found to be higher in the case of first morning voids or 24-hour urine samples, neither of which are available for this particular environmental NHANES sample [18, 63]. While exposure misclassification is possible, it is unlikely to be differential by mortality status. We were also unable to control for specific dietary patterns (e.g., red meat consumption, grilled/smoked foods consumption) and ambient air pollution; therefore, the potential for residual confounding from non-PAHs components (e.g., $PM_{2.5}$, heavy metals) of these major PAHs sources cannot be ruled out. Further residual confounding by genetic architecture is possible if genetic differences that impact PAH metabolism and levels also impact mortality endpoints. Though we attempted to account for both active and passive smoking in our analysis using a combination of questionnaire (e.g., "Do you now smoke cigarettes?") and smoking biomarker (serum cotinine) data, the available NHANES did not allow for a more refined measure of smoking intensity and duration and thus residual confounding due to smoking is possible and may have biased our results. Another limitation is that we had a relatively short follow-up for longer latency period endpoints such as cancer-specific mortality (from incident cancer). Latency periods for common cancers such as lung and breast are on the order of 10 to 20 years [64]. PAHs are hypothesized to function as both initiators and promoters in carcinogenesis and it is likely our analyses capture more of the latter PAHs effect [48–50]. Moreover, given that we removed individuals with self-reported history of cancer at baseline in cancer mortality analyses, our analysis also predisposed to incident cancers with relatively high mortality (e.g., lung cancer). Additionally, although we had a robust number of individuals with all-cause mortality (N = 934), we had a relatively small number of individuals with cancer mortality (N = 159) and cardiovascular mortality (N = 108). In mixtures analysis, our simultaneous assessment of quartiles of individual PAHs and covariates against relatively sparse cancer and cardiovascular mortality events could have led to overfitting and, therefore, poorer performance of the model.

Our study had several strengths. This is one of first studies on PAHs exposure and overall and cause-specific mortality in a large, representative sample of the U.S. population. Unlike many prior investigations, we were able to establish temporality through critical linkage of two

national databases. NDI is the most comprehensive resource for mortality ascertainment in the U.S.; therefore, the possibility of outcome misclassification in our study is likely minimal. A similar NHANES investigation [33] examined PAH exposure and all-cause mortality in a subset of cycles, though with important distinctions. First, Chen et al. did not account for urinary dilution in PAH measures, which can lead to biased estimates. Secondly, our larger sample size (N = 9,739) compared to Chen et al. (N = 1,409) allows for more robust analyses, including to exploration of effect modification by important sociodemographic and lifestyle factors. Given our study population emanated from a nationally-representative sample, our results may have important public health implications for the U.S. population. In our analyses, we examined the effect of OH-PAHs on mortality in multiple ways, including as a ΣOH-PAHs proxy for PAHs exposure from multiple sources and also as an environmental mixture using quantile g-computation, the latter of which allowed insight into joint OH-PAH contributions.

## Conclusions

Our prospective analysis of the U.S. adult population found ΣOH-PAHs to be associated with higher total and cause-specific mortality, confirming and expanding on some of the prior evidence in a population-based, representative sample. We found evidence of effect modification by smoking status, gender, and race/ethnicity across all-cause, cancer-specific, and cardiovascular-specific mortalities, notably potential disparities for cardiovascular-specific mortality for non-Hispanic black and poorer participants. Quantile g-computation allowed us to further assess the total mixtures effect of OH-PAHs, signaling possible OH-PAH effects that may be of interest in future investigations. Results may inform public health efforts aimed at PAH exposure mitigation.

## Supporting information

**S1 Fig. Schematic diagram of exclusion criteria for study population (NHANES 2001–2015).**
(DOCX)

**S2 Fig. Directed acyclic graph to determine the relationships between hydroxylated PAH metabolites (OH-PAHs) and mortality.**
(DOCX)

**S3 Fig. Weights for urinary OH-PAHs from quantile g-computation, by mortality endpoint.** a) Negative and positive weights for OH-PAHs associated with all-cause mortality. b) Negative and positive weights for OH-PAHs associated with cancer-specific mortality. c) Negative and positive weights for OH-PAHs associated with cardiovascular-specific mortality.
(DOCX)

**S1 Table. Baseline descriptive characteristics in NHANES 2001–2014 by mortality status, prior to exclusion criteria and in the study population.**
(DOCX)

**S2 Table. Pearson correlation coefficients of eight urinary OH-PAHs (ng/L) from participants participating in NHANES 2001–2014 (N = 9739).**
(DOCX)

**S3 Table. Continuous final models of Σ OH-PAHs and all-cause and cause-specific mortality, excluding participants who died within one and two years of baseline.**
(DOCX)

**S4 Table. Associations between creatinine-corrected Σ OH-PAHs (nmol/g Cre) and mortality, including urinary creatinine (g/L) as an additional covariate.**
(DOCX)

**S5 Table. Continuous final models of ΣOH-PAHs and all-cause and cause-specific mortality, not excluding self-reported history of cancer or CVD at baseline.**
(DOCX)

**S6 Table. Multipollutant models examining the association between individual OH-PAHs and all-cause, cancer-specific, and CVD-specific mortality.**
(DOCX)

## Acknowledgments

We thank Alexander Keil for his guidance on the quantile g-computation, and Kyla Taylor, Paige Bommarito, John Bucher, and Scott Masten for their critical review of the manuscript.

## Author Contributions

**Conceptualization:** Achal P. Patel, Suril S. Mehta, Alexandra J. White, Whitney D. Arroyave, Amy Wang, Ruth M. Lunn.

**Data curation:** Achal P. Patel, Suril S. Mehta.

**Formal analysis:** Achal P. Patel, Suril S. Mehta, Nicole M. Niehoff.

**Funding acquisition:** Achal P. Patel.

**Investigation:** Achal P. Patel, Suril S. Mehta.

**Methodology:** Achal P. Patel, Suril S. Mehta, Alexandra J. White, Nicole M. Niehoff.

**Software:** Achal P. Patel, Suril S. Mehta.

**Supervision:** Suril S. Mehta.

**Writing – original draft:** Achal P. Patel, Suril S. Mehta.

**Writing – review & editing:** Achal P. Patel, Suril S. Mehta, Alexandra J. White, Nicole M. Niehoff, Whitney D. Arroyave, Amy Wang, Ruth M. Lunn.

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
