## [Decision Letter · Decision Letter 0]

11 Mar 2021

PONE-D-20-37390

Urinary polycyclic aromatic hydrocarbon metabolites and mortality in the United States: a prospective analysis

PLOS ONE

Dear Dr. Mehta,

Thank you for submitting your manuscript to PLOS ONE. After careful consideration, we feel that it has merit but does not fully meet PLOS ONE’s publication criteria as it currently stands. Therefore, we invite you to submit a revised version of the manuscript that addresses the points raised during the review process.

In particular, reviewers raised some concerns regarding how methods are explained and detailed. These issues are of extreme importance in allowing evaluation and replication of the results by external users, and should be fully addressed. Furthermore, statistical robustness should be better supported and discussed, as pointed out by the reviewers.

We look forward to receiving your revised manuscript.

Kind regards,

Giovanni Signore

Academic Editor

PLOS ONE

Journal Requirements:

Reviewers' comments:

Reviewer's Responses to Questions

**Comments to the Author**

1. Is the manuscript technically sound, and do the data support the conclusions?

Reviewer #1: Partly

Reviewer #2: Yes

2. Has the statistical analysis been performed appropriately and rigorously? 

Reviewer #1: No

Reviewer #2: I Don't Know

3. Have the authors made all data underlying the findings in their manuscript fully available?

Reviewer #1: Yes

Reviewer #2: Yes

4. Is the manuscript presented in an intelligible fashion and written in standard English?

Reviewer #1: Yes

Reviewer #2: Yes

5. Review Comments to the Author

Reviewer #1: Title: Urinary polycyclic aromatic hydrocarbon metabolites and mortality in the United States: a prospective analysis.

The authors report on an analysis using NHANES data to investigate select urinary PAH metabolite concentrations and mortality. Individuals 20 years of age or older for seven consecutive survey cycles between 2001 and 2014 were included in the analysis. In current analysis is comprise of data from 9,739 individuals. Urine samples were analyzed at the National Center for Environmental Health, Centers for Disease Control and Prevention. In total, eight PAH metabolites were quantified, using different methods over the seven survey cycles. Vital status as of December 31, 2015 was ascertained by linkage with the National Death Index. All-cause, cancer-specific, and cardiovascular-specific mortality were reported on. The statistical analyses included Cox proportional hazards regression, using age (years) as the time scale. Covariates were determined a priori with a directed acyclic graph. PAH metabolite mixtures were analyzed with quantile g-computation, a novel statistical approach that has advantages over weighted quantile sum regression. The primary results reported were that the log 10 increase in the sum of Oh-PAH urinary metabolite concentrations were associated with all-cause mortality (HR= 1.39; 95% CI = 1.21-1.61), and possibility for cancer-specific mortality (HR= 1.15; 95 CI = 0.79-1.69) and cardiovascular-specific mortality (HR=1.49; 95% CI = 0.94- 2.33). There was evidence of effect measure modification for age, smoking status, gender and race/ethnicity. Quantile g-computation associations for all-cause (HR=1.15; 95% CI = 1.15; 95% CI = 1.02-1.31), cancer-specific (HR=1.41; 95%CI = 1.05-1.90), and cardiovascular-specific mortality (HR=0.98; 95%CI = 0.66-1.47) were also reported. The authors conclude that total PAH exposure has a role in all-cause and cause-specific mortality and reported that reducing exposure would result in a 11.33% reduction in all-cause mortality, 10.64% reduction in cancer-specific mortality, and a 13.49% reduction in cardiovascular mortality in the United States.

Comments:

Overall, the manuscript is very well written and addresses an important, ubiquitous exposure that at high occupational levels have been demonstrated to cause cancer of the lung, bladder and skin. It is less clear whether lower non-occupational levels are associated with increased risk. Specific comments are detailed below:

1) Neither the introduction nor the discussion section describes the substantial literature published on coke oven workers and mortality. Given that coke oven workers have some of the highest reported exposure to PAHs, the results of these studies seem particularly relevant to the current manuscript, especially the mortality studies. These studies clearly showed association with working on the tops of coke ovens with lung cancer, but increases in cardiovascular were not as clearly demonstrated. These data need to considered and discussed in the manuscript.

2) The methods section indicates that 2,577 excluded from the analysis due to missing information or improbable values for follow-up time or accidental deaths. It seems that 20% of the sample is missing, could this missingness have biased the associations, especially because the observed associations were generally small. In addition, the number of individuals 20 years of age and older who donated a urine samples is not reported. The reader has to calculate it from then text. It would be more useful to the reader to just report that there 12,316 individuals 20 years of age or older with urine samples.

3) Individuals with a history of cancer were excluded from the analysis of cancer-specific mortality as were those with a history of CVD for the cardiovascular-specific mortality analyses. It is customary to removed prevalent cases at baseline from analyses intended to assess incidence of a disease, but extending this rationale to analyses intended to assess mortality seems as if it could bias the hazard ratios. A sensitivity analysis would probably inform on this.

4) The authors indicate that they used a directed acyclic graph (DAG) to a priori determine potential confounders to adjust for in the analysis. This DAG should be presented as a figure either in the manuscript or as a supplemental figure.

5) Smoking is an important source of PAHs and other chemical carcinogens and a well-established lung carcinogen. It this analysis, smoking status was used to adjust for potential confounding. However, this is a crude assessment of smoking behavior and the potential for residual confounding seems a likely limitation. In addition, the results stratified by smoking status with residual confounding. For instance, in the nonsmoker/no ETS stratum, neither cancer-specific nor CVD-specific hazard ratios are associated with PAH levels and the association for all-cause mortality is substantially attenuated. This nonsmoker stratum is likely to be have little to no confounding by smoking. Its only in the active smokers that association are observed and this may be due to residual confounding. The NHANES data are limited in that more detailed and precise measure of smoking are not available, so adequate control of smoking is a potential issue. If this is the case, them many, if not all, the hazard ratios reported that adjust for smoking status may be biased towards strengthening the association.

6) The NHANES survey were designed to recruit a representative sample of the US population, resulting in a wide age distribution of potentially eligible individuals. As such, there may be a delayed entry problem and the regression analyses should use left truncation to avoid bias that might be introduced by delayed entry.

7) There were 934 deaths. Of which, 108 (11.5%) were CVD-specific and 159 (17%) were cancer-specific deaths. According to the CDC, however, the vast majority of deaths in the US are due to CVD and cancer. It does not appear as if this sample of individuals from NHANES is representative of the US population.

8) Percent Attributable Risk (PAR) due to exposure are presented (table 6) for all-cause, cancer-specific, and CVD-specific mortality. PARs assume that there is a causal established and that confounding is adequately controlled to have meaningful interpretations. Given that residual confounding by smoking has not been adequately addressed in this report, that assumption seems to be violated. In addition, all-cause, cancer-specific, and CVD-specific mortality are all measures of aggregate disease processes, and many specific diseases within each aggregate may not be caused by PAHs. This information should be dropped from the manuscript.

Reviewer #2: The authors used urinary metabolite data available from the NHANES and linked the data with a mortality database (NDI) to prospectively evaluate the risk for mortality based on PAH exposures. This was a grand endeavor and one of the few studies I know that sought to link such large databases. The research is novel, with the one exception noted below, and the statistical analyses appear robust, although I will acknowledge that some of the analyses are outside my knowledge area, and I am not familiar with their strengths and limitations. Therefore, I do recommend that a biostatistician examine the statistical models.

The discussion and conclusions are supported by the data and the authors appropriately acknowledge the limitations of the study. Overall, I recommend publication of the manuscript after considering the following revisions:

1. It seems like this research is very similar to a paper published in 2020 by Chen et al. The authors acknowledge this but point out that the Chen et al. study was limited by not accounting for urine dilution in their statistical analyses. I agree, urine dilution is an important factor to consider in the analyses. However, it’s still worth noting in this manuscript how the outcomes of the two studies compared. Did they observe similar associations or not? In the current study, the authors acknowledge that urine adjustments for creatinine, or using creatinine as an adjustment in the model did not impact the results. But what happens if you leave it out of the model completely, like Chen et al did. Are the results similar? This really should be addressed.

2. It would be helpful if the authors could edit some of their terminology and lexicon to suit a more general audience. Some of the phrases and terms I believe are very specific to the field of epidemiology/statistical science. For example, the authors use the term “heterogeneity” in many places throughout the manuscript. I assume they refer to differences in significance or model outputs based on either stratified analyses, or univariate analyses, but it’s not always clear to me. For example, on page 13, the first sentence of the last paragraph says “….substantial heterogeneity was observed by smoking and ae for all-cause mortality, by gender…”. Can you use more general terms than heterogeneity to help readers understand your meaning a bit better?

3. I recommend that the authors consider adding a column all the way to the left in Tables 1 and 2 that includes the p-value describing differences between the categories (alive vs deceased; and sum PAHs by category, respectively). I understand that p-values should not be emphasized as much as the coefficients and beta estimates, but all the same, it helps readers understand which variables are significantly different within a univariate analyses.

4. An effect modifier that appears to be missing from the analysis is stress. Stress can obviously impact health outcomes, including mortality. Is there any data available within NHANES that can be paired to the urinary metabolite levels and dataset to consider the impact of stress within these models? For example, did NHANES measure cortisol? Or is there survey data that reflects measures of social stress?

5. The references need to be edited. They are not in alphabetic order and it was hard to find the references that were cited in the manuscript. In addition, some references are included twice. For example, Burstyn et al. 2003 is found on page 23 and again on page 26. Please edit the references to remove duplicates and alphabetize.

6. A major limitation of this research is the use of spot urine samples. The authors acknowledge this limitation in the discussion, but it would be helpful to clarify the variability that could be expected from relying on spot urine samples. For example, please comment on the magnitude of intra class correlation coefficients that have been reported in urine for PAH metabolites over various time frames (over a day, week, month, etc). Furthermore, the PAH metabolites measured can be linked to multiple potential parent PAH compounds, making it difficult to know the exact PAH that contributed to the metabolite.

7. I’m curious to know how easy it is to match NHANES data with a large database like the NDI. And I’m sure others may have similar thoughts. This cannot have been easy. This would also require accessing personal identifiers from each person that participated in NHANES to track them through the NDI. Was special permission requested and granted to access personal information? Who did the linkage exactly? Can you clarify in the methods section?

8. The methods section indicates that all OH-PAHs were log-transformed prior to statistical analyses. Did the authors confirm that the values were normally distributed after log transformation? If they are not normal after log transformation, please include this information in the paper.

9. Is there any suggestion in the NHANES data that PAH exposures have decreased between 2001-2015? Are the geometric means increasing or decreasing with year of sample collection? Can you comment on this in the manuscript?

10. I believe there is older literature supporting a link between PAH exposure and scrotum cancer in chimney cleaners. The authors may wish to add this to their discussion as they only cite one paper on PAH exposures and lung cancer/smoking.

11. The authors reference the fact that PAHs are lipophilic on page 17, and suggest that tissue specific adiposity and BMI may somehow mediate these associations. This is a concern for lipophilic compounds that have long half-lives in the body (like PCBs, DDE, etc); however, PAHs are rapidly metabolized, and the metabolites are not lipophilic. I think this reference and discussion point should be removed, unless the authors can cite some papers showing an increase in PAH biomarkers following weight loss.

6. PLOS authors have the option to publish the peer review history of their article (what does this mean?). If published, this will include your full peer review and any attached files.

Reviewer #1: No

Reviewer #2: No

---

## [Author Response · Author response to Decision Letter 0]

23 Apr 2021

Thank you for your very helpful comments, it has greatly improved our manuscript. Please see our response to comments document, which details in red our responses and provides tables for easy viewing. Thank you. 

Response to Comments

Reviewer 1

Overall, the manuscript is very well written and addresses an important, ubiquitous exposure that at high occupational levels have been demonstrated to cause cancer of the lung, bladder and skin. It is less clear whether lower non-occupational levels are associated with increased risk. Specific comments are detailed below:

1) Neither the introduction nor the discussion section describes the substantial literature published on coke oven workers and mortality. Given that coke oven workers have some of the highest reported exposure to PAHs, the results of these studies seem particularly relevant to the current manuscript, especially the mortality studies. These studies clearly showed association with working on the tops of coke ovens with lung cancer, but increases in cardiovascular were not as clearly demonstrated. These data need to considered and discussed in the manuscript.

RESPONSE: We have expanded our introduction to include the literature on associations between high levels of PAH exposure through occupations (coke oven work, aluminum production, asphalt work etc.) and both respiratory (malignant, non-malignant) and cardiovascular mortality. 

“Occupations with high levels of PAH exposure, including coke ovens, aluminum production, asphalt, and chimney sweeping are associated with excess mortality from lung and other cancers, cardiovascular diseases, and non-malignant respiratory diseases (Redmond 1983, Bertrand et al. 1987, Burstyn et al. 2003, Bursytn et al. 2005, Hogstedt et al. 2013, Miller et al. 2013, Vimercati et al. 2020).” 

2) The methods section indicates that 2,577 excluded from the analysis due to missing information or improbable values for follow-up time or accidental deaths. It seems that 20% of the sample is missing, could this missingness have biased the associations, especially because the observed associations were generally small. In addition, the number of individuals 20 years of age and older who donated a urine samples is not reported. The reader has to calculate it from then text. It would be more useful to the reader to just report that there 12,316 individuals 20 years of age or older with urine samples.

RESPONSE: As part of our data analysis, we compared baseline characteristics in the study population prior to application of exclusion criteria (e.g., missingness) to baseline characteristics in the final (analytic) study population. Please see the table below. Compared to the study population prior to application of exclusion criteria, there was virtually no difference in terms of baseline characteristics in the final (analytic) population. Based on your comment, we have decided to include this table as a supplement, describe this in our Methods, and added this additional sentence to the Results section “We did not find any differences in baseline characteristics prior to and after application of exclusion criteria (Supplemental Table S1).”

Supplemental Table S1. (see Response to comments document)

Additionally, per your suggestion, we have amended language in the Methods section to indicate only those 20+ years old had available urine samples.

“There were 12,316 individuals at least 20 years old with spot urine sample OH-PAHs biomarker data across the 2001-2014 survey cycles.” 

3) Individuals with a history of cancer were excluded from the analysis of cancer-specific mortality as were those with a history of CVD for the cardiovascular-specific mortality analyses. It is customary to removed prevalent cases at baseline from analyses intended to assess incidence of a disease, but extending this rationale to analyses intended to assess mortality seems as if it could bias the hazard ratios. A sensitivity analysis would probably inform on this.

RESPONSE: We have included an additional supplemental table below which does not exclude individuals with a self-reported history of cancer or CVD at baseline. Comparing to Table 3 in the manuscript, the median concentration of ΣOH-PAHs (nmol/L) does not change. When including individuals with a self-reported history of cancer or CVD at baseline, the adjusted risk estimates for ΣOH-PAHs increase slightly for cancer-specific mortality and decrease slightly for CVD-specific mortality, while remaining non-significant and elevated. Under quartile categorization of ΣOH-PAHs, P-trend for cancer-specific mortality was virtually identical and P-trend for CVD-specific mortality slightly higher when including individuals with a self-reported history of cancer or CVD at baseline. 

We have expanded our manuscript to include this additional sensitivity analysis (see Methods, Results, and Supplemental Tables sections). See additions below.

Methods: “Lastly, we conducted analyses including prevalent cases of CVD and cancer (i.e. self-reported history of CVD or cancer at baseline) to assess the impact exclusion of such cases may have had on our risk estimates.”

Results: “When including baseline cases, the adjusted risk estimates slightly increased for cancer-specific mortality, and decreased for CVD-specific mortality, but remained non-significant and elevated (Supplemental Table S5).”

Supplemental Table is displayed in our Response to comments document.

4) The authors indicate that they used a directed acyclic graph (DAG) to a priori determine potential confounders to adjust for in the analysis. This DAG should be presented as a figure either in the manuscript or as a supplemental figure.

RESPONSE: We have included a directed acyclic graph as Supplemental Figure S2. 

5) Smoking is an important source of PAHs and other chemical carcinogens and a well-established lung carcinogen. It this analysis, smoking status was used to adjust for potential confounding. However, this is a crude assessment of smoking behavior and the potential for residual confounding seems a likely limitation. In addition, the results stratified by smoking status with residual confounding. For instance, in the nonsmoker/no ETS stratum, neither cancer-specific nor CVD-specific hazard ratios are associated with PAH levels and the association for all-cause mortality is substantially attenuated. This nonsmoker stratum is likely to be have little to no confounding by smoking. Its only in the active smokers that association are observed and this may be due to residual confounding. The NHANES data are limited in that more detailed and precise measure of smoking are not available, so adequate control of smoking is a potential issue. If this is the case, them many, if not all, the hazard ratios reported that adjust for smoking status may be biased towards strengthening the association.

RESPONSE: Thank you for raising this point. We agree that residual confounding by smoking is possible given that data for confounding control were limited by availability of variables in NHANES. 

We attempted to account for both active and passive smoking by using both reported smoking and a biomarker for tobacco exposure. Specifically, we used a combination of answers to three questions (Q1: “Smoked at least 100 cigarettes in life”; Q2: “Age started smoking cigarettes regularly”; Q3: “Do you now smoke cigarettes”) as well as serum cotinine levels to determine smoking status. We also highlight the complexities of interpreting findings for the combination of tobacco smoke and PAH exposure in our discussion of our smoking-stratified analysis: “Findings by smoking status in relation to all-cause mortality highlighted that PAH-related mortality risk from non-tobacco smoke sources is substantial and that PAHs from tobacco smoke further elevate mortality risk, possibly through differential PAHs profile and/or synergy among PAHs and other deleterious components of tobacco smoke such as carbon monoxide and formaldehyde.”

We recognize the broader limitation that confounding control could be further improved by better availability of data on smoking intensity (e.g., cigarettes per week) and duration (e.g., pack years). As such, we have amended language in the discussion to reflect this and also added further detail regarding construction of the smoking variable in the methods. Please see our language below.

Methods: “Individuals were categorized as current active smokers or ETS exposed based on self-report (Q1: “Smoked at least 100 cigarettes in life?”; Q2: “Age started smoking cigarettes regularly”; Q3: “Do you now smoke cigarettes?”) or serum cotinine concentrations >10 ng/mL.”

Discussion: “Though we attempted to account for both active and passive smoking in our analysis using a combination of questionnaire (e.g., “Do you now smoke cigarettes?”) and smoking biomarker (serum cotinine) data, the available NHANES did not allow for a more refined measures of smoking intensity and duration, and thus, residual confounding due to smoking is possible and may have biased our results.”

6) The NHANES survey were designed to recruit a representative sample of the US population, resulting in a wide age distribution of potentially eligible individuals. As such, there may be a delayed entry problem and the regression analyses should use left truncation to avoid bias that might be introduced by delayed entry.

RESPONSE: In our analyses, we use age as the time scale, where age at NHANES interview determines study entry (start of follow-up) and age where individuals either died or were censored determines study exit (end of follow-up). Such an approach addresses potential confounding by age on the PAH exposure and mortality relationship while also accounting for left-truncation/delayed entry (by comparison, if time since interview were used as the time scale in analyses, there would be bias due to left-truncation/delayed entry because those entering into study at more recent NHANES survey cycles would have to have survived up until those later study entry times) (Lamarca et al. 1998). We additionally control for potential temporal/period effects through adjustment for survey cycle in multivariable models.

Lamarca R, Alonso J, Gómez G, Muñoz A. Left-truncated data with age as time scale: an alternative for survival analysis in the elderly population. J Gerontol A Biol Sci Med Sci. 1998 Sep;53(5):M337-43. doi: 10.1093/gerona/53a.5.m337. PMID: 9754138.

7) There were 934 deaths. Of which, 108 (11.5%) were CVD-specific and 159 (17%) were cancer-specific deaths. According to the CDC, however, the vast majority of deaths in the US are due to CVD and cancer. It does not appear as if this sample of individuals from NHANES is representative of the US population.

RESPONSE: Thank you for making this point. As we highlight in the manuscript, we intended to capture mortality from incident cancers and CVD for those cause-specific analyses. To this end, in analyses for those endpoints, we excluded individuals with self-reported history of cancer or CVD (for each endpoint, respectively). Prior to this exclusion, we had 222 cancer-specific deaths and 173 CVD-specific deaths, which amounts to ~42% of the mortality events (~ 24% for cancer and ~ 19% for CVD). In 2007 (midpoint of follow-up in this study), cancer and CVD accounted for 49% of mortality events, although mortality from CVD was higher than that for cancer.

Therefore, our analytic sample for all-cause mortality analyses, which did not feature exclusions based on self-reported history of either cancer or CVD (N=934), is largely representative of the US population. However, we recognize that the analytic samples for the CVD and cancer-specific endpoints are not as representative and have amended language pertaining to discussion of results for the cause-specific endpoints accordingly. 

8) Percent Attributable Risk (PAR) due to exposure are presented (table 6) for all-cause, cancer-specific, and CVD-specific mortality. PARs assume that there is a causal established and that confounding is adequately controlled to have meaningful interpretations. Given that residual confounding by smoking has not been adequately addressed in this report, that assumption seems to be violated. In addition, all-cause, cancer-specific, and CVD-specific mortality are all measures of aggregate disease processes, and many specific diseases within each aggregate may not be caused by PAHs. This information should be dropped from the manuscript.

RESPONSE: Based on the reviewer’s comments, we agree that the PAR analysis should be dropped. We have deleted all mention in the manuscript. 

Reviewer #2: The authors used urinary metabolite data available from the NHANES and linked the data with a mortality database (NDI) to prospectively evaluate the risk for mortality based on PAH exposures. This was a grand endeavor and one of the few studies I know that sought to link such large databases. The research is novel, with the one exception noted below, and the statistical analyses appear robust, although I will acknowledge that some of the analyses are outside my knowledge area, and I am not familiar with their strengths and limitations. Therefore, I do recommend that a biostatistician examine the statistical models.

The discussion and conclusions are supported by the data and the authors appropriately acknowledge the limitations of the study. Overall, I recommend publication of the manuscript after considering the following revisions:

1. It seems like this research is very similar to a paper published in 2020 by Chen et al. The authors acknowledge this but point out that the Chen et al. study was limited by not accounting for urine dilution in their statistical analyses. I agree, urine dilution is an important factor to consider in the analyses. However, it’s still worth noting in this manuscript how the outcomes of the two studies compared. Did they observe similar associations or not? In the current study, the authors acknowledge that urine adjustments for creatinine, or using creatinine as an adjustment in the model did not impact the results. But what happens if you leave it out of the model completely, like Chen et al did. Are the results similar? This really should be addressed.

RESPONSE: We agree that accounting for urine dilution prior to conducting statistical analysis is critical, as the results would be substantially biased by inter-individual variability in urine volume. As Chen et al. did not account for urinary dilution, our results are not comparable in this sense, and a presentation of unadjusted estimates would be biased. 

Apart from accounting for urine dilution in multiple ways (creatinine adjustment, creatinine correction), our study populations are also different. We expanded our population to include four additional survey cycles (corresponding to years 2007-2014 compared to Chen et al. 2001-2006), resulting in substantially higher mortality events. We also adopted a different analytic approach, where we examined total PAHs exposure and performed a mixtures analysis compared to single PAH exposure estimates (without correction for the dense correlation structure across single PAHs) presented in Chen et al. Therefore, direct comparability of results across these two analyses is difficult. However, we have added language regarding broad comparisons to Chen et al. while expanding on differences between the studies to highlight that they aren’t directly comparable. 

“To date, only one study has examined this relationship using a limited sample of the NHANES population (Chen et al. 2020). This study did not account for urinary dilution of PAH concentrations and the number of mortality endpoints was sparse, resulting in highly imprecise estimates for several PAHs. Furthermore, Chen et al. did not account for the dense correlation structure between PAH metabolites.”

“Our findings for all-cause mortality and cardiovascular-specific mortality support an emerging body of epidemiologic literature on the health impact of PAHs, including Chen et al. (although we note that the studies are not directly comparable due to lack of urinary dilution, differences in study population size, and analytic approach).”

“With regards to gender-specific differences in cancer outcomes in relation to PAH exposure in this study as well as in Chen et al. (2010), Guo et al. (2014) suggested that women are more susceptible than men to chromosomal damage and oxidative stress consequences of PAH (�[NAP, FLUO, PHEN, PYR]) exposure (Guo et al. 2014).”

2. It would be helpful if the authors could edit some of their terminology and lexicon to suit a more general audience. Some of the phrases and terms I believe are very specific to the field of epidemiology/statistical science. For example, the authors use the term “heterogeneity” in many places throughout the manuscript. I assume they refer to differences in significance or model outputs based on either stratified analyses, or univariate analyses, but it’s not always clear to me. For example, on page 13, the first sentence of the last paragraph says “….substantial heterogeneity was observed by smoking and ae for all-cause mortality, by gender…”. Can you use more general terms than heterogeneity to help readers understand your meaning a bit better?

RESPONSE: The term “heterogeneity” refers to differences seen between results in our stratified analysis. We have gone through the manuscript to use more general terms (e.g., “differences between”) to help provide greater clarity in our phrasing. 

3. I recommend that the authors consider adding a column all the way to the left in Tables 1 and 2 that includes the p-value describing differences between the categories (alive vs deceased; and sum PAHs by category, respectively). I understand that p-values should not be emphasized as much as the coefficients and beta estimates, but all the same, it helps readers understand which variables are significantly different within a univariate analyses.

RESPONSE: As we mention in the methods, we used a directed acyclic graph (DAG) approach for identification of a minimally-sufficient adjustment set for confounding control. Statistical significance of covariates in relation to the exposure and outcome variables in univariate analyses alone is likely to be biased given that this is an observational study. We do not believe it would be appropriate to make statistical inferences from crude differences, and given that substantive knowledge (i.e., prior literature) rather than statistical significance in univariate analyses was the criterion for identification of variables for confounding control, we err on the side of caution and describe differences in our manuscript results in Table 1 and 2. This decision to not include p-values on Table 1 has also been supported by multiple publications, including the STROBE guidelines (Hayes-Larson et al. 2019, Vandenbroucke et al. 2007). 

Hayes-Larson E, Kezios KL, Mooney SJ, Lovasi G. Who is in this study, anyway? Guidelines for a useful Table 1. J Clin Epidemiol. 2019 Oct;114:125-132. 

Vandenbroucke JP, von Elm E, Altman DG, Gøtzsche PC, Mulrow CD, Pocock SJ, Poole C, Schlesselman JJ, Egger M; STROBE Initiative. Strengthening the Reporting of Observational Studies in Epidemiology (STROBE): explanation and elaboration. Epidemiology. 2007 Nov;18(6):805-35.

4. An effect modifier that appears to be missing from the analysis is stress. Stress can obviously impact health outcomes, including mortality. Is there any data available within NHANES that can be paired to the urinary metabolite levels and dataset to consider the impact of stress within these models? For example, did NHANES measure cortisol? Or is there survey data that reflects measures of social stress?

RESPONSE: Thank you for this comment. Cortisol is not measured by NHANES, though measures of oxidative stress (e.g., IL-6) and allostatic load are available. Two variables we evaluate as effect modifiers (family poverty status, race/ethnicity) are well-established and strong predictors of allostatic load so we suspect findings stratified by allostatic load would mirror existing findings by family poverty status, for example. A more refined investigation of social stress with robust variables such as cortisol would be interesting for future investigations but beyond the scope of our research question. 

5. The references need to be edited. They are not in alphabetic order and it was hard to find the references that were cited in the manuscript. In addition, some references are included twice. For example, Burstyn et al. 2003 is found on page 23 and again on page 26. Please edit the references to remove duplicates and alphabetize.

RESPONSE: Thank you. We have fixed references to be in alphabetical order, and have removed duplicates. 

6. A major limitation of this research is the use of spot urine samples. The authors acknowledge this limitation in the discussion, but it would be helpful to clarify the variability that could be expected from relying on spot urine samples. For example, please comment on the magnitude of intra class correlation coefficients that have been reported in urine for PAH metabolites over various time frames (over a day, week, month, etc). Furthermore, the PAH metabolites measured can be linked to multiple potential parent PAH compounds, making it difficult to know the exact PAH that contributed to the metabolite.

RESPONSE: Thank you, we have expanded on this in the discussion and included estimates of intraclass correlation coefficients. 

“Biomonitoring studies demonstrate moderate temporal reliability of spot urinary measurements for PAHs; Dobraca et al. (2018) reported intraclass correlation coefficients (ICC) ranging from 0.07 to 0.53 across PAH metabolites considered in this study and most PAH metabolites ranged between 0.35 and 0.50 across multiple years. Importantly, concentrations from a single spot sample reliably ranked exposure into quartiles consistent with study’s multi-year average (Dobraca et al. 2018). Li et al. (2010) reported an ICC of 0.55 for 1-PYR. Reliability was found to be higher in the case of first morning voids or 24-hour urine samples, neither of which are available for this particular environmental NHANES sample (Li et al. 2010, Dobraca et al. 2018).”

7. I’m curious to know how easy it is to match NHANES data with a large database like the NDI. And I’m sure others may have similar thoughts. This cannot have been easy. This would also require accessing personal identifiers from each person that participated in NHANES to track them through the NDI. Was special permission requested and granted to access personal information? Who did the linkage exactly? Can you clarify in the methods section?

RESPONSE: This data linkage has existed for some time (https://www.cdc.gov/nchs/data-linkage/mortality.htm), and linkage is done through CDC/NCHS. More information is available here (https://www.cdc.gov/nchs/data/datalinkage/LMF2015_Methodology_Analytic_Considerations.pdf). 

We used all publicly-available files (as noted in the “Mortality outcomes” section of the Methods), though researchers can submit proposals to use restricted-use data, which includes specific tumor sites, etc. We have included additional clarification in the methods section that NCHS provided the publicly-available files. 

“We utilized the publicly available Linked Mortality Files provided by NCHS, updated through December 31, 2015, which served as end of follow-up (i.e., point of administrative censoring) (https://www.cdc.gov/nchs/data-linkage/mortality-public.htm). All NDI data is based on death certificates coded to International Classification of Diseases version 10 codes (ICD-10).”

8. The methods section indicates that all OH-PAHs were log-transformed prior to statistical analyses. Did the authors confirm that the values were normally distributed after log transformation? If they are not normal after log transformation, please include this information in the paper.

RESPONSE: All OH-PAH metabolites were log-normally distributed. Prior to our statistical analysis, we performed analytical checks to confirm that log transformation normalized our data.

“All OH-PAHs were log-transformed following visual and quantitative ascertainment of right skewness. Upon log transformation, all OH-PAHs were normally distributed.”

9. Is there any suggestion in the NHANES data that PAH exposures have decreased between 2001-2015? Are the geometric means increasing or decreasing with year of sample collection? Can you comment on this in the manuscript?

RESPONSE: We have included language relevant to these trends and disparities in the discussion.

“Across survey cycles (2001-2014), PAHs exposure was highest in 2005-2006, although no clear trends were apparent”.

“In the US, non-Hispanic black and individuals below the poverty level have higher cardiovascular mortality compared to other race/ethnicities and individuals above the poverty level (Van Dyke et al. 2018). We also saw overall higher OH-PAH concentrations among these minority population groups, indicating higher overall PAH exposure. Trend analysis from non-smoking NHANES participants from 2001-2014 also revealed increasing concentrations of PAHs among minority groups, including Non-Hispanic blacks and Mexican Americans (Hudson-Hanley et al. 2021).”

10. I believe there is older literature supporting a link between PAH exposure and scrotum cancer in chimney cleaners. The authors may wish to add this to their discussion as they only cite one paper on PAH exposures and lung cancer/smoking.

RESPONSE: Based on comments by both reviewers, we have expanded our Introduction to include the literature on associations between high levels of PAH exposure through occupations (coke oven work, aluminum production, asphalt work etc.) and respiratory (malignant, non-malignant) and cardiovascular mortality.

“Occupations with high levels of PAH exposure, including coke ovens, aluminum production, asphalt, and chimney sweeping are associated with excess mortality from lung and other cancers, cardiovascular diseases, and non-malignant respiratory diseases (Redmond 1983, Bertrand et al. 1987, Burstyn et al. 2003, Bursytn et al. 2005, Hogstedt et al. 2013, Miller et al. 2013, Vimercati et al. 2020).” 

11. The authors reference the fact that PAHs are lipophilic on page 17, and suggest that tissue specific adiposity and BMI may somehow mediate these associations. This is a concern for lipophilic compounds that have long half-lives in the body (like PCBs, DDE, etc); however, PAHs are rapidly metabolized, and the metabolites are not lipophilic. I think this reference and discussion point should be removed, unless the authors can cite some papers showing an increase in PAH biomarkers following weight loss.

RESPONSE: We have removed mention of this in the manuscript.

---

## [Decision Letter · Decision Letter 1]

21 May 2021

Urinary polycyclic aromatic hydrocarbon metabolites and mortality in the United States: a prospective analysis

PONE-D-20-37390R1

Dear Dr. Mehta,

We’re pleased to inform you that your manuscript has been judged scientifically suitable for publication and will be formally accepted for publication once it meets all outstanding technical requirements.

Kind regards,

Giovanni Signore

Academic Editor

PLOS ONE

Additional Editor Comments (optional):

Reviewers' comments:

Reviewer's Responses to Questions

**Comments to the Author**

1. If the authors have adequately addressed your comments raised in a previous round of review and you feel that this manuscript is now acceptable for publication, you may indicate that here to bypass the “Comments to the Author” section, enter your conflict of interest statement in the “Confidential to Editor” section, and submit your "Accept" recommendation.

Reviewer #1: All comments have been addressed

Reviewer #2: All comments have been addressed

2. Is the manuscript technically sound, and do the data support the conclusions?

Reviewer #1: (No Response)

Reviewer #2: Yes

3. Has the statistical analysis been performed appropriately and rigorously? 

Reviewer #1: (No Response)

Reviewer #2: I Don't Know

4. Have the authors made all data underlying the findings in their manuscript fully available?

Reviewer #1: (No Response)

Reviewer #2: Yes

5. Is the manuscript presented in an intelligible fashion and written in standard English?

Reviewer #1: (No Response)

Reviewer #2: Yes

6. Review Comments to the Author

Reviewer #1: (No Response)

Reviewer #2: I am not a statistician and cannot comment on the rigor of the statistics. All my comments have been addressed, with the exception of comment #3. I still think the p-value would be of use in these tables; however, as I am not a statistician, I will go with the judgement of the editor and Reviewer #1.

7. PLOS authors have the option to publish the peer review history of their article (what does this mean?). If published, this will include your full peer review and any attached files.

Reviewer #1: No

Reviewer #2: No

---

## [Editor Report · Acceptance letter]

27 May 2021

PONE-D-20-37390R1 

Urinary polycyclic aromatic hydrocarbon metabolites and mortality in the United States: a prospective analysis 

Dear Dr. Mehta:

I'm pleased to inform you that your manuscript has been deemed suitable for publication in PLOS ONE. Congratulations! Your manuscript is now with our production department. 

Kind regards, 

on behalf of

Dr. Giovanni Signore 

Academic Editor

PLOS ONE